# Radiofrequency Ablation in Vertebral Body Metastasis with and without Percutaneous Cement Augmentation: A Systematic Review Addressing the Need for SPINE Stability Evaluation

**DOI:** 10.3390/diagnostics13061164

**Published:** 2023-03-18

**Authors:** Stefano Colonna, Andrea Bianconi, Fabio Cofano, Alessandro Prior, Giuseppe Di Perna, Giuseppe Palmieri, Gianluigi Zona, Diego Garbossa, Pietro Fiaschi

**Affiliations:** 1Section of Neurosurgery, Department of Neuroscience, AOU Città della Salute e della Scienza, University of Turin, Corso Bramante 88/90, 10126 Turin, Italy; 2Unità di Chirurgia Vertebrale, Humanitas Gradenigo Hospital, 10100 Turin, Italy; 3Section of Neurosurgery, Department of Neuroscienze, Riabilitazione, Oftalmologia, Genetica e Scienze Materno-Infantili, IRCCS Policlinico San Martino, University of Geneva, Largo Rosanna Benzi, 10, 16132 Genova, Italy; 4Unità di Chirurgia Vertebrale, Casa di Cura Città di Bra, 12042 Cuneo, Italy

**Keywords:** vertebral metastases, spine stability, radiofrequency ablation, thermal ablation, vertebroplasty, kyphoplasty

## Abstract

Vertebral body metastases (VBM) are one of the most frequent sites of bone metastasis, and their adequate therapeutic management still represents an insidious challenge for both oncologists and surgeons. A possible alternative treatment for VBM is radiofrequency ablation (RFA), a percutaneous technique in which an alternating current is delivered to the tumor lesion producing local heating and consequent necrosis. However, RFA alone could alter the biomechanics and microanatomy of the vertebral body, thus increasing the risk of post-procedure vertebral fractures and spine instability, and indeed the aim of the present study is to investigate the effects of RFA on spine stability. A systematic review according to PRISMA-P guidelines was performed, and 17 papers were selected for the systematic review. The results show how RFA is an effective, safe, and feasible alternative to conventional radiotherapy for the treatment of VBM without indication for surgery, but spine stability is a major issue in this context. Although exerting undeniable benefits on pain control and local tumor recurrence, RFA alone increases the risk of spine instability and consequent vertebral body fractures and collapses. Concomitant safe and feasible therapeutic strategies such as percutaneous vertebroplasty and kyphoplasty have shown synergic positive effects on back pain and improvement in spine stability.

## 1. Introduction

Vertebral body metastases (VBM) are one of the most frequent sites of bone metastasis, representing up to approximately 90% of spinal masses found on imaging in oncological patients [1]. Thoracic vertebrae are the most common localization of VBM, accounting for 70% of the cases, followed by lumbosacral (22%) and cervical vertebrae (8%), mainly deriving from hematogenous spreads in the Batson’s vertebral venous plexus. VBM represent a secondary cause of death in the US with a median overall survival after surgery of 8.5 months, with colon, breast, prostate, thyroid, renal cell, lung, and skin cancers as the main primitives [2,3]. Clinical presentation is heterogeneous, with symptoms varying from back pain and functional limitation to metastatic spinal cord compression (MSCC) with potentially permanent neurological deficit resulting from collapse or fracture of the affected vertebral body [1]. The pathogenesis of back pain in VBM is multifactorial and still the object of debate, resulting from a combination of vertebral body instability, local release from tumor cells of pro-inflammatory cytokines with osteoclast activity augmentation, and involvement of periosteal nerve endings [2].

The therapeutic management of VBM is still debated. Spinal metastases are frequently a sign of advanced oncological pathology, often resulting in palliative treatments with the primary aim of increasing life expectancy and improving pain and quality of life. Radiotherapy (RT) is the actual gold standard treatment of VBM in cases without instability and/or spinal cord compression; nevertheless, it presents limitations in terms of latency between treatment and pain relief, presence of radio-resistant tumors, and high recurrence of pain with limited re-treatment availability [4]. An alternative treatment for VBM is radiofrequency ablation (RFA), a percutaneous technique in which an alternating current is delivered in the tumor lesion producing local heating and consequent necrosis while preserving healthy adjacent tissues. Several studies have proven the feasibility and efficacy of RFA in the treatment of painful VBM. However, RFA alone could alter the biomechanics and microanatomy of the vertebral body, thus increasing the risk of post-procedure vertebral fractures and spine instability. Concomitant vertebral reinforcement procedures such as percutaneous vertebroplasty (PVP) and percutaneous kyphoplasty (PKP) are meant to support ablative techniques as RFA, providing stabilization of the vertebral body and restoring vertebral height [2].

The aim of the present study is to investigate the effects of RFA on spine stability and biomechanics after treatment of VBM.

## 2. Materials and Methods

A systematic review according to PRISMA-P (Preferred Reporting Items for Systematic review and Meta-Analysis Protocols) guidelines was performed (Figure 1). We used an online database search (Medline/Pubmed) applying the following research terms used as free terms, keywords, or MeSH terms: “spinal”, “metastases”, “radiofrequency”, “thermal ablation” “stability”, “vertebral bone”, and combining them with AND, OR, NOT operators. The selection process was characterized by the following inclusion criteria: (1) availability of the manuscript in English or an English translation, (2) primary clinical or preclinical studies investigating the use of radiofrequency ablation in vertebral bone metastases with a focus on post-treatment stability, and (3) adult population.

In order to include as many relevant articles as possible, there were no restrictions on the date of publication. Titles and abstracts from the search results page were independently screened for eligible studies by three review authors (SC, PF, AB). Disagreements were resolved through consensus by discussion with a fourth senior author (FC). The following exclusion criteria were applied: (1) papers that mentioned tumors other than vertebral metastases, (2) papers that compared stability after surgical treatment, (3) papers where radiofrequency ablative treatment was associated with surgical treatment, and (4) case reports. Reviews and meta-analyses were cited in the discussion but not considered in the systematic review.

## 3. Results

The first research retrieved a total of 284 papers: from the initial results page we selected clinical and pre-clinical studies and excluded case reports and reviews. Duplicates and non-English language papers were removed. After screening of titles and abstracts, 30 articles were selected for full text reading; we also screened the reference lists in order to identify further relevant papers. Finally, 17 papers were selected for the systematic review.

The first table (Table 1) summarizes the main characteristics of the studies included in the review. Of the 17 papers considered, there were 11 retrospective studies, 1 single-center prospective study, 2 pilot studies, 1 single-arm prospective multicenter study, 1 single-center experience, and 1 cadaveric simulation study. A total of 780 patients were overall considered in the selected studies. There was significant heterogeneity regarding the type of procedures performed among the studies. A total of four studies considered concomitant RFA plus PVP/PKP, one study considered plasma-mediated RFA plus PVP/PKP, three studies compared RFA alone with RFA plus PVP/PKP, two studies compared PVP alone with RFA plus PVP/PKP, one study evaluated RFA, microwave ablation (MWA), cryoablation (CA) plus PVP and adjuvant radiotherapy, two studies considered RFA plus PVP plus concomitant posterior open or percutaneous transpedicular fixation, two studies considered RFA plus PVP plus adjuvant radiotherapy, one study compared PVP plus RFA with PVP plus 123-Iodine radiation therapy, and one study compared PVP plus RFA, 123-Iodine radiation therapy, standard radiation therapy, or zoledronic acid. Study endpoints focused mainly on evaluation of post-procedural pain, quality of life, and spinal stability. Pain was estimated with the Numeric Rating Scale (NRS-11), Visual Analogue Scale (VAS), and Modified Oswestry Low Back Pain Disability Index (MODI). Quality of life was evaluated with Eastern Cooperative Oncology Group Performance Status scale (ECOG-PS), Functional Assessment of Cancer Therapy—General scale (FACT-G7) and Functional Assessment of Cancer Therapy—Bone Pain scale (FACT-BP). One study considered post-procedural neurological evaluation through Frankel classification. Only one study evaluated local tumor recurrence with contrast-enhanced MRI or FDG-PET. Biomechanical stability and spinal stenosis were respectively evaluated with load-induced canal narrowing score (LICN) and MRI spinal stenosis rate (SSR). In cases where PVP/PKP was performed, the Saliou filling score was used to evaluate volume and distribution pattern of cement.

The second table (Table 2) summarizes the results of the selected studies. In almost all cases, the procedure lasted no longer than 60 min, with most procedures lasting less than 15 min per level. Conscious sedation and local anesthesia were the preferred type of anesthesia. All procedures were conducted under CT or fluoroscopy guidance, with non-enhanced CT scan as the favored post-procedural imaging exam. In four studies, no post-procedural complications were described. Almost all complications were either asymptomatic or transient. Post-RFA complications included local edema, numbness of lower extremities, transient aggravation of lower extremity function, abnormal stool function, and abnormal urine function after the operation and new onset of neuropathic pain. In only one case asymptomatic somatic vertebral fracture after RFA alone was described. Post-PVP complications mainly included paravertebral, venous, cortical, epidural or neural foramina bone cement extravasation. In only one case asymptomatic intervertebral disk rupture was described. Post-procedural follow-up protocols were highly heterogeneous, varying from a minimum of 3 days to a maximum of 48 months. In most cases, local tumoral recurrence was not evaluated. Nevertheless, in four studies partial tumor progression was described.

### Limitations of the Study

In the reviewed studies there is no standardization of assessment of post-procedural spinal stability, duration of follow-up is highly variable, and finally, absence of randomized controlled trials comparing ablative strategies alone or associated with vertebroplasty or vertebral fixation is a major issue. The lack of homogeneity in the assessment of instability does not allow for a qualitative judgement supported by statistical analysis.

## 4. Discussion

According to present data in the literature, adequate therapeutic management of VBM still represents an insidious challenge for both oncologists and surgeons. The current National Institute for Health and Care Excellence (NICE) guidelines on spinal metastases suggest analgesia, radiotherapy (RT), bisphosphonates, vertebral cement augmentation, and surgery as the main therapeutic approaches in patients with painful VBM without MSCC [14]. Surgical options must be tailored on the specific clinical case, varying from open or endoscope-assisted vertebrectomy or anterior corpectomy to posterior open or percutaneous pedicle screw stabilization and bone cement augmentation [19,20]. The choice of the surgical approach depends both on the source of compression and on the type of decompression of the circumference of spinal cord [21]. Moreover, circumferential decompression of the spinal cord and nerve roots as in separation surgery provides preservation or restoration of neurological function and enables complementary adjuvant treatments as stereotactic radiosurgery (SRS) or stereotactic body radiation therapy (SBRT) [22]. In patients who are not strictly candidates for spine surgery, less invasive treatment approaches are often preferred in order to reduce the recovery period and lower the morbidity and mortality associated to the therapy.

### 4.1. Overview on Non-RFA Therapeutic Strategies

Radiation therapy as conventional fractionated external beam RT (EBRT) or hypofractionated stereotactic body radiotherapy (SBRT) represents the actual standard treatment of VBM without MSCC. Variable latency between treatment and benefits on biological back pain has been described, usually reaching its maximal effects after 12 to 20 weeks after the procedure. As already mentioned, limitations related to EBRT protocols include failure in local control for radio-resistant tumors, relatively high percentages of pain recurrence after treatment, and limited retreatment opportunities. Moreover, risk of adverse events such as radiation-induced myelopathy and vertebral compression fractures has been reported in 1–5% and 11–39% of the cases, respectively [4,8]. Hormone therapy (HT) and chemotherapy (CT) are possible strategies limited to specific primary tumors such as breast cancer and prostate cancer, in particular in cases with diffuse and recurrent systemic disease. Despite being applicable in specific and selected cases, neither HT nor CT are associated to acceptable pain relief in patients with VBM. Bisphosphonates such as zoledronic acid have shown some benefit in the treatment of metastatic bone pain when associated with RFA, mainly due to their antineoplastic and antiosteoclastic activity in the tumor environment; nevertheless, when compared to other combination strategies such as RFA and bone cement augmentation, they provide a lower grade of pain relief and higher risk of local recurrence [12].

### 4.2. Radiofrequency Ablation

Radiofrequency ablation (RFA) is an effective, safe, and feasible alternative to conventional RT for the treatment of VBM without indication for surgery. Several studies have already demonstrated its benefits in terms of pain reduction in either benign or malignant spinal oncological lesions. Proschek et al. [6] in 2009 evaluated the effects of thermal ablation alone on eight patients with VBM demonstrating improvement of both VAS score and Quality of Life Oswestry index score after the treatment, respectively in 100% and 48.4% of the cases, with no major complication nor evidence of local recurrence at follow-up. In 2019, Sayed et al. [3] treated 30 VBM patients with RFA alone showing consistent improvement of NRS-11 score and FACT-G7 after treatment in all patients, underlining the significantly more rapid effects on pain reduction of RFA compared to standard external beam RT. The underlying mechanisms of pain reduction are still debated. Disruption of periosteal sensory nerve fibers and debulking of the tumor mass, avoiding nerve fiber transmission and stimulation, respectively, seem to represent the main cornerstones of its effectiveness in pain control [12]. Advantages of RFA include low rates of complications, short procedural duration, and reduced patient discomfort. The rapid effects on pain relief after RFA make this technique particularly suitable for patients with short life expectancies. Moreover, RFA is a repeatable technique and can be associated with other treatment strategies such as RT itself or cement bone augmentation [4]. Treatment of lesions close to the spinal cord and nerves can be challenging due to iatrogenic thermal damage. Technological advancement such as in plasma-mediated RFA (pmRFA) and accurate selection of inclusion criteria and temperature settings could help limiting the risk of iatrogenic nervous damage. It is recommended to perform RFA on spinal lesions at least 1 cm distant to vital structures and in the presence of intact cortical bone as a safety margin due to its decreased heat transmission compared with soft tissues. Georgy et al. in 2009 [7] performed 44 pmRF ablations on 37 patients with a combination of vertebral cortical disruption, epidural extension, and paraspinal extension, reporting pain relief in 89% of the cases after a 2–4-week follow-up. Cement extravasation was observed in 73% of the cases, although asymptomatic in almost all the cases except for only one case with temporary radicular pain. While being effective and safe on pain relief, when performed alone RFA has shown a tendency to create or worsen spine stability. Debulking of the tumor creates a cavity in the vertebral body, thus altering the micro anatomy of adjacent healthy osseous trabeculae and the axial and radial force distribution, leading to posterior vertebral wall instability and increased risk of burst fractures or vertebral collapses [14].

### 4.3. Cement Augmentation Techniques

Cement augmentation techniques such as PVP and PKP in the treatment of VBM rely on the necessity to enhance adequate vertebral body stability after ablative therapies such as RFA. Cement injection in the vertebral body such as polymethyl methacrylate (PMMA) helps to preserve mechanical stability and height of vertebral body after the creation of bone cavities during ablative treatments. Additionally, cement augmentation techniques have shown benefits on pain reduction with multifactorial mechanisms. Trabecular stabilization together with exothermic reaction and local chemical toxicity from PMMA lead to adjacent periosteal nerve reduced activity and reduction of mechanical back pain [10,12]. The feasibility of PVP/PKP alone in the treatment of VBM has not been clearly evaluated. Despite the already proven benefits on pain control and spine stability, there is still insufficient data regarding antitumoral effects of PVP. Yang et al. in 2011 hypothesized that the antitumoral effects of PMMA injection could be the result of a cytotoxic and microvascular ischemic effect secondary to exothermic reactions of the cement in the bone [23,24]. Nevertheless, lack of RCT studies comparing PVP and RFA antineoplastic effects and tumoral recurrency rates impedes to formulate adequate indications regarding the use of PVP/PKP alone in VBM treatment. Moreover, without any preceding tumor mass ablation procedures, the presence of heterogenous pathological tissue in the context of the vertebral body could alter the distribution of cement, resulting in suboptimal effects on pain relief and spine stabilization [9]. Bone cement leakage is a common complication in PVP/PKP, reported in 4.8–39% of the cases, although in most case asymptomatic except for less common epidural or intracanalar extravasation, which could present with spinal cord or neural compression symptoms, eventually requiring surgical treatment [1,10].

### 4.4. Combination of Ablative and Cement Augmentation Treatments and Spine Stability

Spinal stability, as well as pain control, is a primary element to consider in the evaluation of therapeutic strategies for VBM. Preliminary assessment of spinal stability is crucial to establish whether the treatment strategy must be surgical or conservative. Spinal instability neoplastic score (SINS) and the Neurology-Stability-Epidural compression (NSE) score are reliable scoring tools that can help to identify patients who may benefit from surgical intervention, thus representing valid instruments in the primary evaluation of the treatment strategy [25,26,27,28]. Nonetheless, assessment of spinal stability is equally fundamental after conservative ablative treatments such as RFA. Several studies have already proven the synergic effects of concomitant RFA and cement augmentation techniques such as PVP/PKP on pain control in patients with VBM. In 2004, Masala et al. [5] treated three VBM patients with a combination of RFA and cement augmentation, demonstrating a consistent improvement of post-procedural VAS score in all the patients included, with an average decrease in 6 VAS scale points after the treatment. Similarly, Reyes et al. [15] in 2017 treated 72 VBM lesions in 49 patients with combination therapy, showing a mean decrease in VAS score and ODI score after the treatment of 4.6 and 13.4 points, respectively. In addition, the concomitant association of RFA and vertebroplasty appears to improve also post-procedural spine stability. Several mechanisms have been proposed to clarify the synergic effect of the combination therapy. The complementarity of ablative and reconstructive approaches seems to rely on a balance between both desirable and undesirable effects of each treatment. RFA alone produces a bone cavity in the vertebral body, with both benefits on local tumor control and pain relief and disadvantages in terms of spine stability and risk of post-procedural collapses. Vertebroplasty alone helps to restore local mechanical stability of the spine, although with the potential risk of a suboptimal effect caused by the presence of the tumoral mass interfering with cement distribution and potentially facilitating cement leakage. With the combination approach, the bone cavity obtained from the debulking of the tumoral mass allows to improve the distribution rather than the volume of cement injected, thus resulting in optimal vertebral body strengthening and stabilization. Moreover, microthrombosis of peritumoral venous vascularization obtained with thermal damage from RFA helps to reduce the risk of venous leakage of bone cement during PKP [12]. Combination therapy shows additional crucial benefits in terms of spine stability when compared to RFA or vertebroplasty alone. One of the potential limitations of VBM treatment with RFA and vertebroplasty are lesions in the posterior aspect of the vertebral body, often associated with posterior cortical wall damage and consequent instability of the entire vertebra with the risk of intracanalar involvement. In compromised posterior cortical walls, the altered isolating effect of the damaged cortical bone can facilitate thermal damage of intracanalar nervous structures. Moreover, with cement augmentation techniques alone, a compromised posterior wall increases the risk of epidural cement leakage. With combination therapy, in particular with advanced ablative techniques such as plasma mediated RFA (pmRFA), precise ablation of posterior lesions followed by finely controlled cement injection manages to obtain optimal local tumor control and adequate vertebral stabilization [7].

According to the available literature, concomitant association of RFA and cement augmentation techniques can improve mechanical stability of the spine, especially if cement can be distributed in order to adequately support the posterior vertebral body wall, decreasing significantly the risk of post-procedural vertebral fractures.

## 5. Conclusions

Spine stability after ablative therapies such as radiofrequency ablation is a major issue in the context of VBM treatment. Although exerting undeniable benefits on pain control and local tumor recurrence, RFA alone increases the risk of spine instability and consequent vertebral body fractures and collapses. Concomitant safe and feasible therapeutic strategies such as PVP and PKP have shown synergic positive effect on back pain and improvement of spine stability, especially in case of adequate posterior vertebral body stabilization and efficient distribution of cement in the context of the lesion. Further studies should be conducted in order to help clarify the effect of both ablative and reconstructive therapies on spine stability and biomechanics in the treatment of VBM.

## Figures and Tables

**Figure 1 diagnostics-13-01164-f001:**
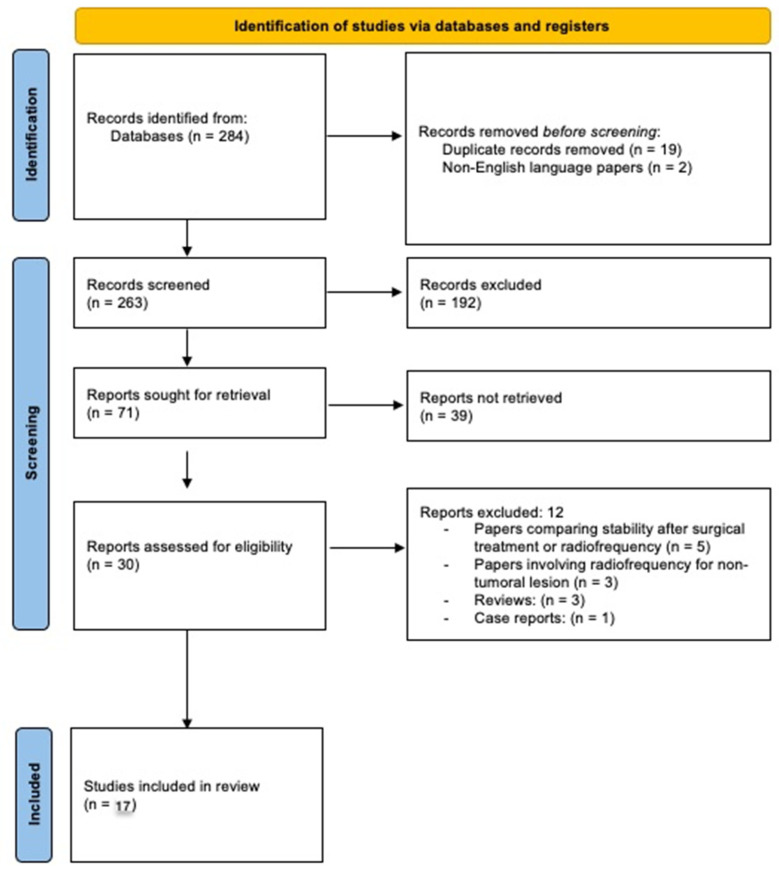
PRISMA flow chart for literature search.

**Table 1 diagnostics-13-01164-t001:** Summary of characteristics of included studies.

Authors and Year	Study Location	Type of Study	No. of Patients (No. of Lesions)	Type of Procedure	Inclusion Criteria	Exclusion Criteria	Study Endpoints
Sayed et al., 2019 [3]	USA	Single-center prospective study	30 (34)	RFA (2) vs. RFA + PVP (28)	At least one painful thoracic or lumbar metastasis, age at least 18 years	Vertebral metastatic disease in the cervical spine, spinal cord compression from posterior tumor extension	Evaluation of pain reduction (NRS-11); improvement of quality of life (FACT-G7)
Masala et al., 2004 [5]	Italy	Single-center experience	3	RFA + PVP	Tokuhashi prognostic scoring system < 6	Osteoblastic tumors, retropulsion fractures, spread of tumor within the epidural space, local infection, coagulative disorders, involvement or missing integrity of pedicles or joint facets	Evaluation of pain reduction (VAS)
Proschek et al., 2009 [6]	Germany	Pilot study	16	RFA (8) vs. RFA + PVP (8)	Mechanical back pain, absence of neurological deficit	Vertebral fractures, radicular neurological symptoms, coagulation disorders, local infection	Evaluation of pain reduction (VAS); improvement of quality of life (Oswestry Disability Questionnaire)
Arrigoni et al., 2020 [4]	Italy	Pilot study	11	RFA, microwave ablation (MWA), cryoablation (CA) + PVP + adjuvant RT	At least one vertebral osteolytic metastatic lesion, disabling refractory back pain, KPS > 70	Asymptomatic lesions, spinal osteoblastic metastatic lesions without risk of fracture, platelets count < 50,000, local or systemic infection	Stability of the vertebral lesion 6 months after treatment (RECIST criteria); improvement of pain (VAS); improvement of quality of life (ECOG-PS)
Georgy et al., 2009 [7]	USA	Retrospective study	37	pmRFA + PVP	Painful vertebral metastasis associated with at least one of the following criteria: cortical disruption, epidural extension, paraspinal extension	Painful vertebral metastasis not associated with at least one of the following criteria: cortical disruption, epidural extension, paraspinal extension	Evaluation of cement disposition pattern (standard PACS System); evaluation of pain reduction (VAS)
Bagla et al., 2016 [8]	USA	Single-arm prospective multicenter study	50	RFA + PVP	Painful vertebral bone metastasis in at least one thoraco-lumbar vertebra, age at least 18 years, pain concordant to the metastatic lesion site	Painful vertebral bone metastasis in cervical spine, posterior tumor extension with cord compression	Improvement of pain (NPRS); improvement of back-related disability (MODI); improvement of quality of life (FACT-G7, FACT-BP)
Giammalva et al., 2022 [2]	Italy	Retrospective study	54	Concomitant RFA + PVP + posterior open/percutaneous transpedicular fixation	Karnofsky score ≥ 60, unremitting thoraco-lumbar pain (VAS score ≥ 5), osteolytic lesion on neuroimaging, unresecable tumors (according to Tokuhashi score), intractable pain with chemotherapy, radiation therapy and refractory to analgesic drugs	Karnofsky score < 60, mild thoraco-lumbar pain (VAS < 5), osteoblastic tumors on neuroimaging, general contraindications for surgery (infection, allergy, bleeding diseases), intradural and intramedullary tumors and neurological impairments caused by spinal metastasis itself	Evaluation of pain improvement (VAS); bone distribution of cement (Saliou filling score)
He et al., 2021 [9]	China	Retrospective study	40	PVP + RFA (19) vs. PVP + ^123^I (21)	Clear history and pathological diagnosis of malignant tumors; improved CT and enhanced MRI findings of the spine before the operation; narrowing of the spinal canal and epidural compression in the local spinal cord visible on sagittal MRI scans before surgery; osteolytic bone destruction of spinal metastasis; one or several clinical manifestations of the following secondary spinal cord injury: (a) local or radiation pain and progressive aggravation; (b) sensory function damage and progressive aggravation; (c) motor function damage and progressive aggravation; (d) sphincter function abnormality; and (e) involvement of < 4 vertebral body segments in the tumors	Predicted survival time of < 3 months; primary spinal tumors, such as multiple myeloma; spinal infectious diseases, such as spinal tuberculosis and other bacterial infections; severe cardiovascular and cerebrovascular diseases, respiratory failure, liver and kidney failure, and inability to tolerate surgery; coagulation dysfunction; severe skin infection in the operation area	Evaluation of pain improvement (VAS); evaluation of spinal stenosis rate on MRI (SSR)
Lane et al., 2010 [10]	Canada	Retrospective study	36 (53)	RFA + PVP	Focal pain clinically localized to a region with imaging confirming the presence of bony tumor involvement; pain partially or totally refractory to analgesic medications; unacceptable side-effects of additional medication; at least 18 years of age; life expectancy of greater than 1 month. Bony metastases with adjacent soft tissue and/or posterior aspect of vertebral body tumor involvement were not excluded.	Extensive pathological destruction of the posterior wall of the vertebral body with > 40% reduction in the antero-posterior canal dimension; purely osteosclerotic metastases; patients with INRs > 1.3; platelet counts < 50,000; and any local or systemic infection	Evaluation of pain improvement (VAS)
Lu et al., 2017 [11]	China	Retrospective study	169	PVP + RFA (51) PVP + ^123^I (49) PVP + zoledronic acid (38) PVP + RT (31)	Not specified	Not specified	Evaluation of pain improvement (VAS; WHO Pain Relief scale); evaluation of motor dysfunction (ODI)
Lv et al., 2020 [12]	China	Retrospective study	87 (125) Group A: 35 (47) Group B: 52 (78)	PVP + RFA (Group A) vs. PVP (Group B)	Definite diagnosis of spinal metastatic cancer (pathological or cytological diagnosis); structurally intact posterior margin of the vertebral body without nerve root symptoms; thoracic and lumbar vertebral body lesions, which are mainly lesions of osteolytic destruction or mixed destruction; willingness to undergo the proposed procedure (signed informed consent) and relatively good treatment compliance	Incomplete structure of the posterior margin of the vertebral cortex or infiltration of tumor into the dura, accompanied by nerve root symptoms; osteogenic lesions; terminal patients; severe cardiopulmonary disease or coagulation dysfunction	Evaluation of pain and function improvement (VAS, ODI); anterior and intermediate vertebral body height; bone cement leakage; local tumor recurrence
Maugeri et al., 2017 [13]	Italy	Retrospective study	18	RFA + PVP + posterior transpedicular fixation	Unresecable tumors, according to Tokuhashi score; Karnofsky score > 60; osteolytic lesion on neuroimaging; VAS > 5; intractable pain with CT, RT or other treatments	General contraindications for surgery (infection, allergy, bleeding diseases); poor general condition (Karnofsky score < 60); osteoblastic tumors on neuroimaging; VAS < 5; spinal cord or nerve compression or intradural and intramedullary tumors	Evaluation of pain improvement (VAS); bone distribution of polymethylmethacrylate (Saliou filling score)
Pezeshki et al., 2016 [14]	Canada	Cadaver simulation study	6 (cadaver specimens)	RFA; RFA + PVP	/	/	Evaluation of the biomechanics of the spine after RFA + PVP (load-induced canal narrowing, LICN)
Reyes et al., 2017 [15]	USA; Italy	Retrospective multicenter study	49 (72)	RFA + PVP	Not specified	Not specified	Evaluation of pain and function improvement (VAS; ODI); evaluation of local tumor recurrence (contrast-enhanced MRI, FDG-PET)
Abdelgawaad et al., 2021 [16]	Germany; Egypt	Retrospective study	60 (75)	RFA + BKP + adjuvant RT	Painful osteolytic spinal metastases refractory to analgesics; absence of neurologic deficit or cord compression; stable posterior column according to the Spinal Instability Neoplastic Score (SINS). Posterior vertebral wall defects were not considered contraindications.	Entirely osteoblastic lesions; lesions associated with marked spinal instability requiring spinal instrumentation surgery	Evaluation of pain improvement (VAS)
Wallace et al., 2015 [17]	USA	Retrospective study	72 (110)	RFA + PVP (95%) ± adjuvant RT	Not specified	Entirely osteoblastic lesions; lesions associated with pathologic compression fracture with spinal instability or causing metastatic spinal cord compression. Tumor within 1 cm of the spinal cord or nerves was not a contraindication for RFA.	Evaluation of pain improvement (NRS-11)
Yang et al., 2017 [18]	China	Retrospective study	42 (52)	RFA + PVP (25); PVP alone (17)	Evident history of the primary tumor, or diagnosis by aspiration biopsy; experienced sudden or persistent pain in the neck, chest, back, or waist; complete clinical record available	Not specified	Evaluation of pain improvement (VAS); evaluation of neurological status (Frankel classification)

**Table 2 diagnostics-13-01164-t002:** Specifics of included studies.

Authors and Year	Mean Age (Range)	Type of Anesthesia	Procedure Modality	Post-Procedural Imaging	Average Duration of Procedure	Post-Procedure Complications	Post-Procedure Follow-Up Duration	Local Recurrence During Follow-Up
Sayed et al., 2019 [3]	62.9 ± 13.45	Conscious sedation; local anesthesia	CT/fluoroscopy-guided	Non-enhanced MRI/CT scan	9.56 min	None	3 days, 1 week, 1 month, 3 months	No local recurrence
Masala et al., 2004 [5]	72.3 (63–82)	Conscious sedation; local anesthesia	CT/fluoroscopy-guided	Non-enhanced CT scan	35–45 min	None	None	Not specified
Proschek et al., 2009 [6]	59.5 (52–69)	Conscious sedation; local anesthesia	CT/fluoroscopy-guided	Non-enhanced CT scan	Not specified	None	Average of 20.4 months (range 8–36 months)	No local recurrence
Arrigoni et al., 2020 [4]	62.9 (49–76)	Conscious sedation; local or spinal anesthesia	CT/fluoroscopy-guided	Non-enhanced CT scan	Not specified	None	Every 6 months Average 18 months (range 6–48 months)	Enlargement of lesion’s volume and vertebral fracture at 6 months follow-up in 2 patients
Bagla et al., 2016 [8]	61 (23–83)	Conscious sedation (70%) or general anesthesia (30%)	CT/fluoroscopy-guided	Not reported	6.7 min	Neuropathic pain, syncope, rupture of disk adjacent to the treated vertebra	3 days,1 week, 1 month, 3 months	Not specified
Gerogy et al., 2009 [7]	Not reported	Not reported	CT/fluoroscopy-guided	Non-enhanced CT scan	1.5–6 min	Bone cement extravasation (venous, cortical, epidural, neural foramina)	2–4 weeks	Not specified
Giammalva et al., 2022 [2]	63.44 (34–86)	General anesthesia	CT/fluoroscopy-guided	Non-enhanced CT scan	60.4 min	Perivertebral cement leakage	1 week, 1 month, 3 months, 6 months	Not specified
He et al., 2021 [9]	Median age 58 (18–76)	Local anesthesia	CT/fluoroscopy-guided	Non-enhanced CT scan	10–15 min	Local edema, increased pain, numbness of the lower extremities, transient aggravation of lower extremity function, decreased mobility in some patients after the operation, abnormal stool function and abnormal urine function after the operation	1 week, 1 month, 3 months	Not specified
Lu et al., 2017 [11]	56.9 (37–77)	Conscious sedation; local anesthesia	CT/fluoroscopy-guided	Non-enhanced CT scan	Not specified	Asymptomatic bone cement extravasation (paravertebral soft tissues, paravertebral veins, epidural space, adjacent disk) after PVP + RFA	1 week, 1 month, 6 months	Not specified
Lv et al., 2020 [12]	56.9 (37–77)	Conscious sedation; local anesthesia	CT/fluoroscopy-guided	Non-enhanced CT scan	1.5–2.5 min for anterior vertebral body lesions 2.5–3.5 min for posterior vertebral body lesions	Asymptomatic bone cement leakage (paravertebral soft tissues, epidural space, adjacent disk) Group A: 6.4% Group B: 20.5%	3 days, 1 month, 6 months	Group A: 11.4% (4) Group B: 30.8% (11)
Maugeri et al., 2017 [13]	55.72 (34–69)	General anesthesia	CT/fluoroscopy-guided	Non-enhanced CT scan	60.4 min	Asymptomatic bone cement leakage (lateral spinal recess)	1 week, 1 month, 3 months, 6 months	Not specified
Pezeshki et al., 2016 [14]	Cadaver specimens	Cadaver specimens	CT/fluoroscopy-guided	Non-enhanced CT scan	12 min	/	/	/
Reyes et al., 2017 [15]	64.3 ± 12.6	Conscious sedation; local anesthesia	CT/fluoroscopy-guided	Non-enhanced CT scan (23); Contrast-enhanced MRI (10); Non-enhanced MRI (8); FDG-PET (10)	3.7 ± 2.5 min	None	2–4 weeks	New extension of tumor into a neural foramen with epidural extension (1); increased epidural extension after ablation due to unexpected development of new malignancy (1)
Abdelgawaad et al., 2021 [16]	69 ± 10.2 (50–79)	General anesthesia	CT/fluoroscopy-guided	Plain X-rays	Not specified	Asymptomatic leaks into the needle track (2), into draining veins (2) and into the disk space (1)	3 days, next follow-up at least at 6 months (mean 13.2 ± 6.3)	Not specified
Wallace et al., 2015 [17]	68.4 ± 18.8	Conscious sedation; local anesthesia	CT/fluoroscopy-guided	Not specified	8 min 32 s ± 4 min 49 s	Post-procedure temporary radicular pain (4); 60% percent (3/5) of the radiofrequency ablated vertebrae that were not augmented fractured within the subsequent 12 months.	1 week, 4 weeks	Not specified
Yang et al., 2017 [18]	62.7 ± 9.0 (46–82)	Conscious sedation; local anesthesia	CT/fluoroscopy-guided	Not specified	5–15 min	Asymptomatic cement leakage (2); spinal cord compression symptoms (1) improved after 7-days medical therapy.	1, 3, 6, and 12 months (mean duration 7.8 months)	Partial tumor progression, with the tumors penetrating the nerve segments (5)

## Data Availability

All data generated or analyzed during this study are included in this published article.

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
