# Peer review of "Radiofrequency Ablation in Vertebral Body Metastasis with and without Percutaneous Cement Augmentation: A Systematic Review Addressing the Need for SPINE Stability Evaluation"

_diagnostics, 2023, doi:10.3390/diagnostics13061164_

Round 1

Reviewer 1 Report

The authors address a very important subject and have done a thorough literature research.

In their title and introduction, the authors claim to focus on RFA and spine stability in vertebral body metastasis.  

However, the presentation of the data is very confusing / almost nonexistent

The results section includes a short description of all included papers (pages 5-12, probably supposed to be Table 1), then, in table 2 (pages 13-26) some specifics of the included studies are added.

This table (table 2) is pretty illegible (due to layout, there are every specific is labeled with a number (for no reason?!). The layout of table 2 in the supplementary file is much better (content is the same).

There is no “statistic” or summary of data/results of the studies which would/could give at least some ideas or answers in relation to the title of the presented paper:

For instance:

-       - number of patients (total)

-       - type of tumor(s)

-       - outcome (painscores, quality of life,…)

-       - type of complication(s) (i.e. fracture(s) after RFA (!))

-       - patient survival

-      

Some of those data/results ts are mentioned in the “discussion” part.

The conclusion is purely based on opinion without any meaningful presentation of results to back it up.

Author Response

Dear Reviewer,

We appreciated your suggestions, and we consequently corrected our manuscript. The results section now includes a more precise and detailed description of Table 1 and Table 2. Best layout for our tables to be clearer and more legible should be vertical, due to the number of parameters included in each table. A richer description of results and parameters, along with some further addition to the discussion part, should now fit better with conclusions.

RESULTS

The first research retrieved a total of 284 papers: from the initial results page we selected clinical and pre-clinical studies and excluded case reports and reviews. Duplicates and non-English language papers were removed. After screening of titles and abstracts, 30 articles were selected for full text reading; we also screened the reference lists in order to identify further relevant papers. Finally, 17 papers were selected for thesystematic review.

The first table (Table 1) summarizes the main characteristics of the studies included in the review. Of the 17 papers considered, there were 11 retrospective studies, 1 single-centre prospective studies, 2 pilot studies, 1 single-arm prospective multicentre study, 1 single-centre experience and 1 cadaveric simulation study. A total of 780 patients were overall considered in the selected studies. There was significant heterogeneity regarding the type of procedures performed among the studies. A total of 4 studies considered concomitant RFA plus PVP/PKP, 1 study considered plasma-mediated RFA plus PVP/PKP, 3 studies compared RFA alone with RFA plus PVP/PKP, 2 studies compared PVP alone with RFA plus PVP/PKP, 1 study evaluated RFA, microwave ablation (MWA), cryoablation (CA) plus PVP and adjuvant radiotherapy, 2 studies considered RFA plus PVP plus concomitant posterior open or percutaneous transpedicular fixation, 2 studies considered RFA plus PVP plus adjuvant radiotherapy, 1 study compared PVP plus RFA with PVP plus 123-Iodine radiation therapy, and 1 study compared PVP plus RFA, 123-Iodine radiation therapy, standard radiation therapy or zoledronic acid. Study endpoints focused mainly on evaluation of post-procedural pain, quality of life and spinal stability. Pain was estimated with Numeric Rating Scale (NRS-11), Visual Analogue Scale (VAS) and Modified Oswestry Low Back Pain Disability Index (MODI). Quality of live was evaluated with Eastern Cooperative Oncology Group Performance Status scale (ECOG-PS), Functional Assessment of Cancer Therapy - General scale (FACT-G7) and Functional Assessment of Cancer Therapy – Bone Pain scale (FACT-BP). One study considered post-procedural neurological evaluation through Frankel classification. Only one study evaluated local tumour recurrence with contrast-enhanced MRI or FDG-PET. Biomechanical stability and spinal stenosis were respectively evaluated with Load-induced canal narrowing score (LICN) and MRI spinal stenosis rate (SSR). In cases were PVP/PKP was performed, Saliou filling score was used to evaluate volume and distribution pattern of cement.

The second table (Table 2) summarizes the results of the selected studies. In almost all cases the procedure lasted no longer than 60 minutes, with most procedures lasting less than 15 minutes per level. Conscious sedation and local anaesthesia were the preferred type of anaesthesia. All procedures were conducted under CT or fluoroscopy guidance, with non-enhanced CT scan as the favored post-procedural imaging exam. In 4 studies no post-procedural complications were described. Almost all complications were either asymptomatic or transient. Post-RFA complications included local oedema, numbness of lower extremities, transient aggravation of lower extremity function, abnormal stool function and abnormal urine function after the operation and new onset of neuropathic pain. In only one case asymptomatic somatic vertebral fracture after RFA alone was described. Post-PVP complications mainly included paravertebral, venous, cortical, epidural or neural foramina bone cement extravasation. In only one case asymptomatic intervertebral disk rupture was described. Post-procedural follow-up protocols were highly heterogeneous, varying from a minimum of 3 days to a maximum of 48 months. In most cases local tumoral recurrence was not evaluated. Nevertheless, in 4 studies partial tumour progression was described.

3.1 Limitations of the study

In the reviewed studies there is no standardization of assessment of post-procedural spinal stability andduration of follow-up is highly variable. Moreover, absence of randomized controlled trials comparing ablative strategies alone or associated with vertebroplasty or vertebral fixation is a major issue.

Reviewer 2 Report

The authors provide a comprehensive description of the use and limitations of RFA for VBM. The topic is extremely relevant to the neurosurgical audience, and the analysis is well presented. Please see comments below for minor edits:

In the introduction, there are multiple English language errors. Please have the manuscript revised by an English language editor. The majority of the manuscript is very well written, but there are some minor grammatical mistakes.

Despite the benefits on pain control and spine stability, treatment with cement augmentation techniques alone in VBM is not recommended. Cement injection in the vertebral body has poor to none antineoplastic activity, leading to partial or incomplete destruction of the lesions and thus increasing the risk of tumor recurrence. - Can the authors please provide a justification for this statement? PMMA injection replaces the tumor volume, so it is unclear how this increases the risk of tumor recurrence. Are the authors comparing to RFA alone? We would need to see a citation directly comparing RFA alone to vertebroplasty alone.

Page 23/24 - synergic should be synergistic

The authors refer to the SINS score but I do not see the original citation for the SINS algorithm from the SOSG - please include the original article describing the SINS score.

Author Response

Dear Reviewer,

We appreciated your suggestions, and we consequently corrected our manuscript. 
We corrected some minor English grammar mistakes throughout the manuscript. Moreover, we added references to the original article describing the SINS score by SOSG.
Regarding the discussion, we added a paragraph in the “4.3 Cement augmentation techniques” explaining the plausible relation between PMMA injection and antineoplastic effect, underlying the lack of comparative RCT studies between antineoplastic activity of PVP alone versus RFA alone in the VBM treatment.

"[...] Cement augmentation techniques as PVP and PKP in treatment of VBM rely on the necessity to enhance adequate vertebral body stability after ablative therapies such as RFA. Cement injection in the vertebral body such as polymethyl methacrylate (PMMA) helps to preserve mechanical stability and height of vertebral body after the creation of bone cavities during ablative treatments. Additionally, cement augmentation techniques have shown benefits on pain reduction with multifactorial mechanisms. Trabecular stabilization together with exothermic reaction and local chemical toxicity from PMMA lead to adjacent periosteal nerve reduced activity and reduction of mechanical back pain [10, 12]. 
Feasibility of PVP/PKP alone in the treatment of VBM has not been clearly evaluated. Despite the already proven benefits on pain control and spine stability, there is still insufficient data regarding antitumoral effects of PVP. Hiu-Lin et al in 2011 hypothesized that the antitumoral effects of PMMA injection could be the result of a cytotoxic and microvascular ischemic effect secondary to exothermic reactions of the cement in the bone [New reference 1, New reference 2]. Nevertheless, lack of RCT studies comparing PVP and RFA antineoplastic effects and tumoral recurrency rates impedes to formulate adequate indications regarding the use of PVP/PKP alone in VBM treatment. Moreover, without any preceding tumour mass ablation procedures, the presence of heterogenous pathological tissue in the context of the vertebral body could alter the distribution of cement resulting in suboptimal effects on pain relief and spine stabilization [9]. Bone cement leakage is a common complication in PVP/PKP reported in 4.8-39% of the cases, although in most case asymptomatic except for less common epidural or intracanalar extravasation which could present with spinal cord or neural compression symptoms, eventually requiring surgical treatment [1, 10].

  • [New reference 1] Do vertebroplasty and kyphoplasty have an antitumoral effect?  Hui-Lin Yang, Zhi-Yong Sun, Gui-Zhong Wu, Kang-Wu Chen, Yong Gu, Zhong-Lai Qian Medical Hypotheses Volume 76, Issue 1, January 2011, Pages 145-146 
  • [New reference 2] Balloon kyphoplasty in malignant spinal fractures: a systematic review and meta-analysis Carmen Bouza*, Teresa López-Cuadrado, Patricia Cediel, Zuleika Saz-Parkinson and José María Amate BMC Palliat Care, 8 (2009), p. 12"

Reviewer 3 Report

In this manuscript Colonna et al. report a systematic review on radio frequency ablation in vertebral body metastasis.

The review strictly follows the PRISMA flow chart and it is well structured.

The tables are clear and exhaustive, and the discussion is able to touch all the different aspects of this topic.

I do recommend to check and fix a typo in line 50 where I guess it should be "instability" instead of "stability".

I do not have any further recommendation.

Author Response

Dear Reviewer,
We appreciated your suggestions, and we consequently corrected our manuscript. We corrected the typo in line 50 as you suggested with the word "instability".

Round 2

Reviewer 1 Report

My concerns and comments have only been addressed minimally.

“However, the presentation of the data is very confusing / almost nonexistent”

-       Still confusing

“table 2 is pretty illegible (due to layout(?), every “specific” is labeled with a new number (for no reason?!). The layout of table 2 in the supplementary file is much better (content is the same).

-       Still unchanged

“There is no “statistic” or summary of data/results of the studies which would/could give at least some ideas or answers in relation to the title of the presented paper:

For instance:

-       - number of patients (total)

-       - type of tumor(s)

-       - outcome (painscores, quality of life,…)

-       - type of complication(s) (i.e. fracture(s) after RFA (!))

-       - patient survival

-       …

-       Still no meaningful statistic/summary

-       The authors claim in their abstract that they are presenting a “systematic review”

-       Maybe the author (and publishers) have a completely different view and understanding of “systematic review” (my understanding can be found here (https://en.wikipedia.org/wiki/Systematic_review) and is still not met in the presented paper.

"The conclusion is purely based on opinion without any meaningful presentation of results to back it up"

-       Unchanged.

Author Response

We apologise to the reviewer if he finds that his corrections were approached in a subtle manner, particularly with regard to the table layout, for which it was an unfortunate error.

Nevertheless, we cannot agree that the results are not clearly displayed or even non-existent. There are two tables that summarise the results comprehensively and extensively, and the result section has been expanded in order to report in the main body and not only in the tables the results. If the auditor has more specific doubts about the results, we are more than willing to clarify them.

We apologise for the layout of table 2 and for not editing it, it was a copy-paste error from the original file which we thought we had corrected. The numbers were automatically generated by Microsoft word when the layout was changed to conform to the magazine template

We must also disagree that the review does not qualify as systematic, as the prisma 2020 checklist was strictly followed. In particular, the search query is reproducible and the article selection process is reported in detail. Furthermore, all characteristics of the studies under review are reported in detail. Since this does not qualify as a meta-analysis (and makes no claim to be one), no statistical analysis was conducted.

We agree that a further summary table of the results with more “quantitative” findings, as requested by the reviewer, should be useful, however it was not reported due to the heterogeneity of the studies, in particular with regard to the length of follow-up, but above all the lack of a common assessment of vertebral instability, which makes the results incomparable, as pointed out in the limitations section of the study.

Furthemore also both the outcomes (type of complications) and the preoperative characteristics (eg. Tumor type) reported in the various studies are very heterogeneous and not usefully systematisable.

However, these results were not neglected or not addressed, in fact in the inclusion and exclusion criteria (table 1) the types of tumors treated are shown, as are the complications of the procedures in table 2.

Furthermore, since this is a systematic review, the results of the literature search itself have been described in the results section, while what was reported in the studies themselves has not only been included in the tables in the results section, but has also been dealt with in the discussion section, including 'statistical' elements (e.g. section 4.1 and 4.2)

Conclusions are not based on opinion but on the data in the tables and in the discussion section. As mentioned earlier, the lack of homogeneity in the assesment of instability does not allow for a qualitative judgement supported by statistical analysis, a limitation that has been added in the appropriate section. However, the point that emerged from this review is precisely, as highlighted in the limitations and conclusions, the lack of systematic data on the long-term results of the establishment, which should be homogenized by tumor type, length of follow-up and radiological assessment of instability.

Round 3

Reviewer 1 Report

After the 3rd review of the presented work I conclude the following:

-       I will admit that the authors followed the guidelines for a systematic review and did and do not claim to present a meta analysis (which is not possible due to the heterogeneity of the data) – this is also mentioned in “limitations”

-       Usually, in a systematic review, the quality of the included studies is somehow evaluated – as far as I can see, the authors do mention for each included study the type of study (i.e. “retrospective review”) but not the quality - this may help the audience to more easily and quickly understand  the quality of the data of each included study

-       The title of the work claims to be “focusing on spine stability” – however- as far as I can see- there is only 1 study (Sayed et al 2019) which actually compared RFA alone vs. RFA+PVP (plus one cadaver study (Pezeshiki et al) -  so while the authors may have wanted to focus on spine stability in the context of RFA, the presented work – at least in my view- cannot really answer this question.

-       Some (or even quite a few) of the presented studies may also provide data on the stability of the spine after the intervention – however this is not presented comprehensively neither in one of the tables nor in the text.

-       Overall the authors have put a lot of effort in searching and evaluating the existing literature on RFA in VBM and should be commended for that.

      – in my view, changing the title to i.e.

“Radiofrequency ablation in vertebral body metastasis with and without percutaneous augmentation with cement: a systematic review”

will much more reflect the true content of the paper without promising information/results (evaluation of spine stability in that context) that cannot be fulfilled.

Author Response

We thank the reviewer for the third revision.

It is indeed true that although the authors have tried to focus on spinal stability, the evidence gathered is not sufficiently consistent and the title may be misleading for the audience. Therefore according to the suggestions of the reviewer we changed the title to "Radiofrequency ablation in vertebral body metastasis with and without percutaneous cement augmentation: a systematic review addressing the need for spine stability evaluation"